# Explosive instability due to flow over a rippled bottom

Raunak Raj[1] and Anirban Guha[1,2]

[1]Environmental and Geophysical Fluids Group, Department of Mechanical Engineering, Indian Institute of Technology, Kanpur, U.P. 208016, India.
[2]Institute of Coastal Research, Helmholtz-Zentrum Geesthacht, Geesthacht 21502, Germany.

**Correspondence:** Anirban Guha (anirbanguha.ubc@gmail.com)

**Abstract.** In this paper, we study Bragg resonance, i.e. the triad interaction between surface and/or interfacial waves with bottom ripple, in the presence of background velocity. We show that when one of the constituent waves of the triad has negative energy, the amplitudes of all the waves grow exponentially. This is very different from classic Bragg resonance in which one wave decays to cause the growth of the other. The instabilities we observe are 'explosive' and are different from normal mode shear instabilities since our velocity profiles are linearly stable. Our work may explain the existence of large amplitude internal waves over periodic bottom ripples in the presence of tidal flow observed in oceans and estuaries.

## 1 Introduction

The energy exchange between two counter-propagating surface gravity waves mediated by a bottom ripple, otherwise known as the 'Bragg resonance', is a widely known phenomenon in oceanography and coastal engineering (Davies, 1982; Mei, 1985; Kirby, 1986). Bragg resonance strongly affects the wave spectrum in continental shelves and coastal regions (Ball, 1964), modifies the shore-parallel sandbars, and protects the shoreline from wave attacks (Heathershaw and Davies, 1985; Elgar et al., 2003). Actually, Bragg resonance is a special kind of resonant triad in which one of the constituent waves is the bottom ripple, which acts as a stationary wave (Alam et al., 2009a). Usually, in a classical resonant triad (hereafter, 'resonant' is suppressed for brevity), one wave gives energy to the other two waves via nonlinear interactions such that the individual wave-amplitudes remain bounded at all times.

Not directly related to the problem of wave triad interactions is the concept of 'negative energy waves'. Cairns (1979) showed that, similar to plasma physics, waves possessing negative energy may exist in simple fluid dynamical set-ups. When a negative energy wave is present in a system, increase in its amplitude comes at the cost of decrease in the total energy of the system. Cairns used this concept to explain numerous linear instabilities, where he explained linear instabilities in terms of an interaction between the 'positive energy' and the 'negative energy' branches of the corresponding dispersion curve. Grimshaw (1984) explored negative energy concepts using wave action principles and averaged Lagrangians, which reveals that negative energy waves can only arise in the presence of a background flow. Craik and Adam (1979) connected the theory of negative energy waves to the classical triad interaction problem, in which they theoretically predicted the existence of 'explosive triads'. In an explosive triad, each constituent wave can grow simultaneously while keeping the total energy of the system conserved, making it quite different from the 'usual' triad. As already mentioned, the amplitude of one of the waves in a usual triad

decreases to feed energy into the other two waves; subsequently, these two receiver waves, after reaching their maximum amplitudes, act as donors by transferring energy back to the first wave. When one of the constituent waves of a triad has negative energy and the other two waves have positive energies, then to compensate the energy decrease of the negative energy wave, the positive energy waves have to increase in amplitude, and hence, there is an increase in amplitude of all three waves forming the triad. Explosive growth in the context of Bragg resonance would imply that both of its constituent waves have simultaneous growth while keeping the energy of the system conserved (note that the third 'wave' constituting the Bragg triad is the bottom ripple). We emphasize here that Bragg resonance has been traditionally studied in the absence of any background velocity field, and in such scenarios, explosive growth is impossible.

The present work aims at studying explosive Bragg resonance which occurs as a consequence of the presence of velocity field. We find that contrary to typical Bragg resonances, in explosive Bragg resonance the amplitude of the waves isn't limited by energy considerations. This may explain the presence of large amplitude internal waves over periodic bottom ripples such as those in the Rotterdam waterway (Pietrzak et al., 1990). Such large amplitude response was primarily seen during the strong flood tides and presence of velocity field was deemed essential for the same. Further, this mechanism of wave generation may also explain presence of high amplitude internal waves in continental shelves (Alford et al., 2012) whose amplitudes are uncorrelated with tidal forcing. We also note here that in the presence of a vertically nonuniform current, a work similar to us but for capillary–gravity waves was performed by Voronovich et al. (1980). In the present paper, one of our arguments will be that shear is not necessary for such explosive growth and even a uniform mean flow (flowing over a stationary bottom boundary) is sufficient.

In order to explain the explosive Bragg resonance, we first consider in §2 a single-layered flow. Here, the system consists of a surface wave propagating over a rippled bottom topography and the fluid flowing with a mean velocity profile as shown in figure 1(a). For explosive instability, we expect this wave to grow along with its explosive counterpart while simultaneously conserving the energy of the system. After highlighting the importance of mean flow in explosive Bragg resonance and a detailing various aspects of negative energy waves, we briefly examine a more realistic two-layered flow scenario (figure 1(b)) in §3. Here we explain why internal (interfacial) waves would be more susceptible to such resonances. Continuously stratified flows that can support internal gravity waves have not been studied. We expect explosive Bragg resonance to be a general phenomenon, and can therefore be realized in a variety of systems.

## 2   Theory

A wave of wavenumber $k$ can interact with an undulated bottom of wavenumber $k_b$ to transfer some of its energy to the wavenumbers $k \pm k_b$. However, this energy transfer is maximized when the following resonance condition is satisfied:

$$k_i \pm k_r \pm k_b = 0 \qquad ; \qquad \omega_i \pm \omega_r = 0, \tag{2.1a,b}$$

where $\omega$ denotes frequency, and the subscripts $i$ and $r$ respectively denote incident and resonant waves. For a single layered flow (i.e. having no density variation) in the absence of any mean flow, *only one* such resonant set can exist at the first order

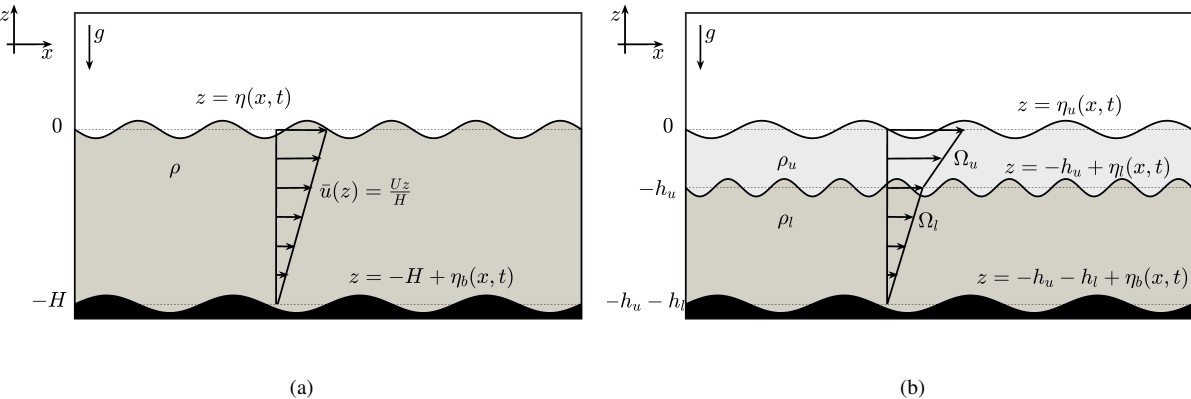

(a)                                                          (b)

**Figure 1.** Schematic diagram showing waves over a rippled bottom. The profile of the bottom is $\eta_b$. (a) One-layered flow of depth $H$ with density $\rho$ is assumed. The free surface profile is $\eta$. The velocity profile (chosen to be linear) must create a Doppler shift between the free surface and the bottom. (b) Two-layered flow of depth $H \equiv h_u + h_l$ with density $\rho_u$ and $\rho_l$ is assumed. The free surface (interface) profile is $\eta_u$ ($\eta_l$). Velocity profile is piecewise linear.

of nonlinearity (Davies, 1982). In this case, an incident surface gravity wave of wavenumber $k_i$ resonantly interacts with the bottom of twice the wavenumber (i.e. $k_b = 2k_i$) to generate exactly one surface gravity wave, which has a wavenumber $k_r = k_i$, and travels in the direction opposite to the incident wave. This is the classic Bragg resonance condition for surface waves. The corresponding dispersion relation is shown in the figure 2(a). Each wave has been vectorially represented as $(k, \omega)$ (marked by arrows) in the dispersion diagram, and the corresponding coordinates (or vector tips) are marked by discs '•'. In this vectorial representation, the 'Bragg triad' forms a vector-triangle (in fact, any resonant triad at the first order of nonlinearity forms a vector triangle in the dispersion diagram). We observe that the dispersion relation for a surface gravity wave is symmetrical about the $k$ axis, i.e. there is no difference between the positively and the negatively traveling waves apart from the direction of propagation.

In the presence of a velocity field, however, the symmetry between the rightward and the leftward traveling waves is destroyed. We find that this not only leads to a modification in the resonance conditions, but also leads to the formation of new resonant triads (Raj and Guha, 2019). We assume a single layered flow with a 'Couette' type mean velocity profile, i.e. the mean velocity increases linearly from $\bar{u} = 0$ at the bottom ($z = -H$) to $\bar{u} = U$ at the surface ($z = 0$); see figure 1(a). The mean shear is denoted by $\Omega \equiv d\bar{u}/dz = U/H$, and is a constant in this situation. We emphasize here that, although the actual velocity profile is of some relevance, what matters the most is the Doppler shift between the bottom ripples and the surface. Therefore, even a uniform current (flowing over a stationary bottom boundary) would have worked just fine. Mathematically, the dispersion relation in this case is given by

$$\omega_{in}^2 + \Omega \tanh(kH)\omega_{in} - gk \tanh(kH) = 0. \tag{2.2}$$

Here $\omega_{in} \equiv \omega - Uk$ is defined as the intrinsic frequency of the wave, which basically means the frequency obtained after subtracting the Doppler shift produced by the mean flow. For this paper, without a loss of generality, we have considered a positive $k$ and have allowed the frequency $\omega$ to be either positive or negative. As a consequence, $\omega > 0$ means a positively travelling wave while $\omega < 0$ means a negatively travelling wave in the stationary reference frame. Equation (2.2), a quadratic equation in $\omega_{in}$, reveals that the product of the two roots is negative. Hence the two branches of the dispersion curves, shown in figure 2(b), have intrinsic frequencies of opposite signs. Note that in figure 2(b), we have plotted $\omega$ vs $k$ (and *not* $\omega_{in}$ vs $k$) in the nondimensional form, and have taken the surface velocity $U > 0$ without loss of generality. The positive intrinsic frequency branch has been labeled as $\mathcal{SG}^+$, while the negative one as $\mathcal{SG}^-$. Below we perform a detailed analysis of the dispersion curve.

## 2.1 An analysis of triads using dispersion curves

From figure 2(b), we see that while $\omega$ for the $\mathcal{SG}^+$ branch increases monotonically with $k$, the same is not true for the $\mathcal{SG}^-$ branch. For this branch, $\omega$ initially decreases with $k$, attains a minima, and then starts to increase. Thus the sign of $\omega$ becomes opposite to $\omega_{in}$; in figure 2(b) this happens after $kH \approx 3$ for $U^* \equiv U/\sqrt{gH} = 0.7$. The non-monotonic behavior of $\omega$ with $k$ allows a given wave-vector $(k, \omega)$ to form *multiple resonant triad sets*. This is starkly different from the classic Bragg resonance, where, as mentioned earlier, only one triad condition is possible for a single layered fluid (figure 2(a)).

The minimum frequency of the $\mathcal{SG}^-$ branch[1] is labeled as $\omega_{min}$. The wavenumber at which $\mathcal{SG}^-$ branch crosses the $k$ axis ($\omega = 0$ line) is labeled as $k_z$. For every point on the dispersion curve with frequency $\omega < |\omega_{min}|$ (shown in shaded region in figure 2(b)), there exists three resonant triads for any given incident wavenumber. For an example, we choose a frequency $\omega_0 < |\omega_{min}|$ and plot $\omega = \pm\omega_0$ on the dispersion curve. There will be four intersections with the dispersion curve, all shown using bullets ($\bullet$) in figure 2(b). An interesting observation here is that three of these above mentioned points lie on the same branch of the dispersion curve $\mathcal{SG}^-$. Any two of the four intersection points form a resonant triad with appropriately chosen bottom ripple. If the points lie on the opposite side of the $k$ axis, the bottom's wavenumber for resonance would be the sum of two wavenumbers involved, else it would be the difference. For a given $|\omega_0|$ less than $|\omega_{min}|$, a total of six ($\equiv {}^4C_2$) Bragg triads can be obtained. We again recall here the in the absence of mean velocity, only one triad would have been possible, as shown in figure 1(a).

If we choose $\omega_0 > -\omega_{min}$, there would be only one such triad between $\mathcal{SG}^+$ branch and $\mathcal{SG}^-$ branch. In that case, since both the incident and the resonant frequencies will be positive, the wavenumber of the bottom ripple would be the difference of the two wavenumbers. However, for the case when $\omega_0 < \omega_{min}$, a stably propagating wave cannot exist and there won't be any chance of resonance.

---

[1]Note that in this case we have taken $\Omega > 0$; had we taken $\Omega < 0$, we would have got a maxima in $\mathcal{SG}^+$ curve rather than getting a minima in $\mathcal{SG}^-$ curve. Basically, the dispersion curve in that case would be a mirror image of figure 2 about the line $\omega = 0$.

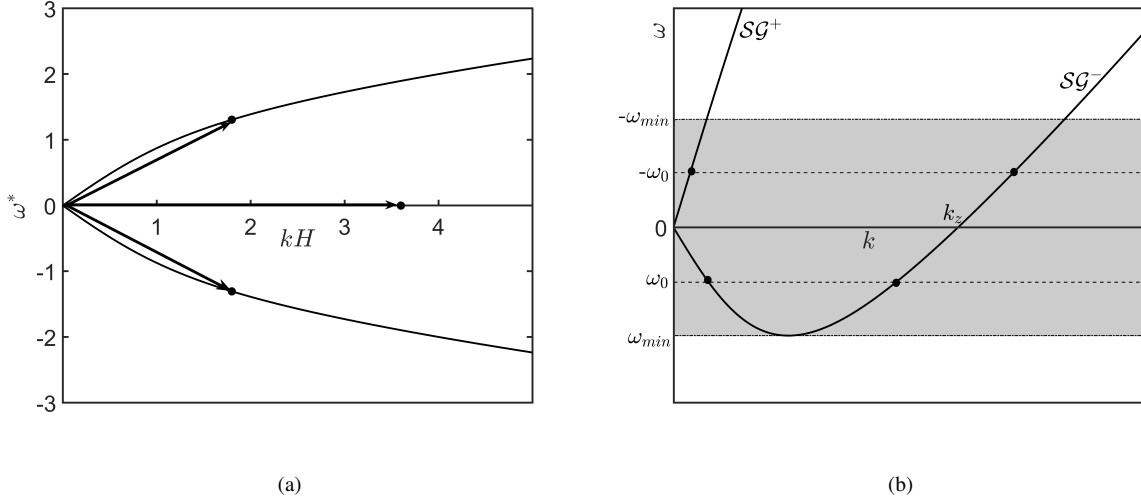

(a)                                    (b)

**Figure 2.** (a) A Bragg triad in the absence of mean velocity. Wavenumber $k$ has been nondimensionalised by $1/H$ ($H \equiv$ flow depth), while frequency $\omega$ by $\sqrt{g/H}$ i.e. $\omega^* \equiv \omega/\sqrt{g/H}$. (b) Bragg triads in the presence of 'Couette-type' mean velocity with $U^* \equiv U/(\sqrt{gH}) = 0.7$. In the shaded region, three Bragg triads can exist for a given $k$. For labels, see text.

## 2.2 Negative energy waves

As shown in Cairns (1979), the energy per unit area $E$ of a wave (hereafter, simply referred to as 'energy') having frequency $\omega$, amplitude $a$, and satisfying the dispersion relation $\mathfrak{D}(\omega, k) = 0$ is given by

$$E = \frac{1}{4}\omega \frac{\partial \mathfrak{D}}{\partial \omega}|a|^2. \tag{2.3}$$

The energy of a wave in the words of Cairns is "the work done during an idealized process in which the wave is driven up by an external force applied on a surface in the fluid." Therefore, a negative energy wave is such a wave whose introduction into the system lowers the energy of the system. For a given system, the dispersion relation $\mathfrak{D}(\omega, k) = 0$ can be written in multiple forms; for every form, the factor $\partial \mathfrak{D}/\partial \omega$ would be different. However, only when the dispersion relation for the purpose of calculating energy is written in the form described in Cairns (1979), we obtain the energy of the wave. In this approach an interface $z = \eta(x, t)$ is chosen, and the pressure $p_1$ ($p_2$) just above (below) this interface is written as

$$p_1(x,t) = \mathfrak{D}_1(\omega, k)\eta(x, t) \quad ; \quad p_2(x,t) = \mathfrak{D}_2(\omega, k)\eta(x, t). \tag{2.4a,b}$$

The dispersion relation is given by

$$\mathfrak{D}(\omega, k) = \mathfrak{D}_1(\omega, k) - \mathfrak{D}_2(\omega, k) \tag{2.5}$$

The dispersion relation of an intermediate depth surface gravity wave in the presence of a constant mean shear current is found to be

$$\mathfrak{D}(\omega,k) = \rho \left[ \frac{(\omega - Uk)^2}{k\tanh(kH)} + \frac{(\omega - Uk)\Omega}{k} - g \right] = 0, \tag{2.6}$$

where $\rho$ is the density of the fluid. Hence,

$$\frac{\partial \mathfrak{D}}{\partial \omega} = \rho \frac{\omega_{in}^2 + gk\tanh(kH)}{\omega_{in} k\tanh(kH)}. \tag{2.7}$$

We observe that in this case, the sign of $\partial\mathfrak{D}/\partial\omega$ is dependent only on the sign of $\omega_{in}$, i.e. the intrinsic frequency of the wave. Therefore, for the $\mathcal{SG}^+$ branch, $\partial\mathfrak{D}/\partial\omega$ is always positive, whereas for the $\mathcal{SG}^-$ branch, it is always negative. Further, for the $\mathcal{SG}^+$ branch, the frequency $\omega$ is always positive. Therefore, according to (2.3), the energy of the $\mathcal{SG}^+$ branch is always positive ($E > 0$). For the $\mathcal{SG}^-$ branch, $\omega < 0$ when $k < k_z$, which implies positive energy for $k < k_z$. However for $k > k_z$, we have

10 $\omega > 0$ but $\partial\mathfrak{D}/\partial\omega$ still remains negative, which implies a negative energy wave ($E < 0$). Therefore, using the negative energy approach of Cairns, we expect that a wave on the branch $\mathcal{SG}^-$, for which $k > k_z$, will form an explosive Bragg triad with a suitable positive energy wave.

## 2.3 An explosive Bragg resonance pair

To derive the amplitude evolution equations for Bragg resonance, we represent the constituent surface waves as $(k_1, \omega_1)$ and

15 $(k_2, \omega_2)$. The waves are expressed in the form $a_1(t)\exp[i(k_1 x - \omega_1 t)] + \text{c.c}$ and $a_2(t)\exp[i(k_2 x - \omega_2 t)] + \text{c.c}$, where c.c denotes complex conjugate. With a stationary bottom ripple having a wavenumber $k_b$ and they satisfy the resonance condition

$$k_1 + k_2 = k_b \quad ; \quad \omega_1 + \omega_2 = 0. \tag{2.8a,b}$$

The amplitude evolution equations for Bragg resonance for such a case is given by (Raj and Guha, 2019):

$$\frac{da_1}{dt} = \beta_1 a_b \bar{a}_2 \quad ; \quad \frac{da_2}{dt} = \beta_2 a_b \bar{a}_1, \tag{2.9a,b}$$

where $a_1$ and $a_2$ are the complex amplitude of the waves involved and $a_b$ is the complex amplitude of the bottom ripple with overbars denoting the complex conjugates. The coefficients $\beta_1$ and $\beta_2$ in (2.9a,b) can be derived following Fredholm's alternative, the procedure of which has been elaborated in Raj and Guha (2019). In this particular case of uniform shear, the coefficients are as follows:

$$\beta_1 = i\frac{k_1(\omega_1 - Uk_1)^2(\omega_2 - Uk_2)}{\cosh k_1 H \sinh k_2 H[gk_1\tanh k_1 H + (\omega_1 - Uk_1)^2]}, \tag{2.10a}$$

$$\beta_2 = i\frac{k_2(\omega_2 - Uk_2)^2(\omega_1 - Uk_1)}{\cosh k_2 H \sinh k_1 H[gk_2\tanh k_2 H + (\omega_2 - Uk_2)^2]}. \tag{2.10b}$$

The coefficients $\beta_1$ and $\beta_2$ are purely imaginary. Furthermore, it can be easily seen that the signs of $\beta_1$ and $\beta_2$ respectively depend on the signs of $\omega_1 - Uk_1$ and $\omega_2 - Uk_2$ only, i.e. only on the intrinsic frequencies of the respective waves. Thus the

sign of the product $\beta_1\beta_2$ depends only on the sign of the product of the intrinsic frequencies of the waves forming the resonant pair. As noted before, the intrinsic frequency of the $\mathcal{SG}^+$ branch is positive for all $k$, while that of the $\mathcal{SG}^-$ branch is negative for all $k$.

We can also express (2.9a,b) as

$$5 \qquad \frac{d^2a_1}{dt^2} = -\beta_1\beta_2|a|_b^2 a_1 \qquad ; \qquad \frac{d^2a_2}{dt^2} = -\beta_2\beta_1|a|_b^2 a_2. \qquad \text{(2.11a,b)}$$

Hence for explosive growth

$$-\beta_1\beta_2 > 0 \Rightarrow (\omega_1 - Uk_1)(\omega_2 - Uk_2) > 0. \qquad \text{(2.12)}$$

This basically means that the intrinsic frequencies have to be of the same sign, or in other words, the two points must lie on the same branch of the dispersion curve in figure 2(b). Furthermore, (2.8a,b) assumes that the actual frequencies of the resonant pair must be of opposite signs. Therefore we deduce that for Bragg resonance to give rise to explosive growth, if the actual frequencies of the waves are of opposite signs, then the intrinsic frequencies must have the same sign. This is precisely the condition satisfied by all the Bragg triads formed by the waves having $0 < \omega < |\omega_{min}|$ on the $\mathcal{SG}^-$ branch with the waves on the same branch having $\omega_{min} < \omega < 0$.

### 2.4 Explosive Bragg resonance from the negative energy perspective

To understand how the waves in §2.3 constitute a positive energy–negative energy pair, we write the coefficients $\beta_1$ and $\beta_2$ in terms of $\partial\mathfrak{D}/\partial\omega$. Using (2.7) along with (2.10), we obtain

$$\beta_1 = \mathrm{i}\lambda\left(\frac{\partial\mathfrak{D}}{\partial\omega}\right)^{-1}_{\omega_1,k_1} \qquad ; \qquad \beta_2 = \mathrm{i}\lambda\left(\frac{\partial\mathfrak{D}}{\partial\omega}\right)^{-1}_{\omega_2,k_2}, \qquad \text{(2.13a,b)}$$

where $\lambda = (\omega_1 - Uk_1)(\omega_2 - Uk_2)[\sinh(k_1 H)\sinh k_2 H]^{-1}$. Therefore (2.9a,b) can be written as

$$\left(\frac{\partial\mathfrak{D}}{\partial\omega}\right)_{\omega_1,k_1}\frac{da_1}{dt} = \mathrm{i}\lambda a_b\bar{a}_2 \qquad ; \qquad \left(\frac{\partial\mathfrak{D}}{\partial\omega}\right)_{\omega_2,k_2}\frac{da_2}{dt} = \mathrm{i}\lambda a_b\bar{a}_1. \qquad \text{(2.14a,b)}$$

Although we have written the expressions in this form for a special case of a single-layered flow, such form is very general. In fact, for the case of triad interactions involving three waves, a similar set of equations was obtained by Craik and Adam (1979). Using (2.13a,b) along with the condition for explosive instability, i.e. (2.12), we get

$$\left(\frac{\partial\mathfrak{D}}{\partial\omega}\right)_{\omega_1,k_1}\left(\frac{\partial\mathfrak{D}}{\partial\omega}\right)_{\omega_2,k_2} > 0. \qquad \text{(2.15)}$$

This basically means that energy coefficients $\partial\mathfrak{D}/\partial\omega$ of the two waves are of the same sign. However, we know from (2.8a,b) that the frequencies are of opposite signs. Hence the energy of the two waves, given by (2.3), must be of opposite signs for an explosive instability to occur.

## 2.5 Another explosive Bragg resonance pair

Here we consider the case where the two frequencies are of the same sign. The resonance condition in this case is given by

$$k_2 - k_1 = k_b \qquad ; \qquad \omega_2 - \omega_1 = 0. \tag{2.16a,b}$$

The amplitude evolution equation is found to be

$$5 \quad \frac{da_1}{dt} = \beta_1 \bar{a}_b a_2 \qquad ; \qquad \frac{da_2}{dt} = \beta_2 a_b a_1, \tag{2.17a,b}$$

where $\beta_1$ and $\beta_2$ remains the same as that in (2.11a,b). The above equations can be written as

$$\frac{d^2 a_1^{(1)}}{dt^2} = \beta_1 \beta_2 |a|_b^2 a_1 \qquad ; \qquad \frac{d^2 a_2}{dt^2} = \beta_2 \beta_1 |a|_b^2 a_2, \tag{2.18a,b}$$

which implies that the condition for explosive growth is

$$\beta_1 \beta_2 > 0 \Rightarrow (\omega_1 - U k_1)(\omega_2 - U k_2) < 0. \tag{2.19}$$

Hence, if the actual frequencies of the waves are of the same sign, the intrinsic frequencies must be of opposite signs for explosive instability to occur. As we have mentioned earlier, having opposite signs of intrinsic frequency in this case means that the waves must be on different branches, i.e. one on $\mathcal{SG}^+$ and the other on $\mathcal{SG}^-$, but have the same sign of actual frequency. As an example, this will correspond to the Bragg triad formed by the two black dots in the upper-half plane (i.e. $\omega > 0$) in figure 2(b). Using (2.13a,b) and (2.19) along with (2.16a,b), we will indeed find that the energy of the two waves must be of opposite
signs for the occurrence of explosive instability.

## 2.6 Generalized analysis

We have already shown the existence of Bragg resonant triads that gives rise to explosive growth for a single layered flow with shear present. We have shown for this case that for explosive instability, the dispersion curve must cross the $\omega = 0$ axis for explosive instability (assuming the reference frame attached to the bottom). Furthermore, if the intrinsic frequency of a
wave is $\omega_{in}$, then for it to change sign, a velocity $U$ in opposite direction must be present, implying that for some value of $k$, $|Uk| > |\omega_{in}|$. Given the fact that frequency of a gravity wave varies as $\omega_{in} \sim k^{1/2}$ and for vorticity wave, $\omega_{in} \sim k^0$, for some value of $k$, $|Uk|$ will exceed $|\omega_{in}|$. Therefore, mathematically, explosive triad will always exist for any value of $U$. However, as can be seen from (2.10), the coefficient $\beta_1$ and $\beta_2$ are proportional to $(\cosh k_1 H \sinh k_2 H)^{-1}$ and $(\sinh k_1 H \cosh k_2 H)^{-1}$ respectively, hence for higher values of $k$, they rapidly tend towards 0. Thus for very low velocities, explosive triad conditions
maybe satisfied for a very large value of $k$ but the growth rate tends to zero in such cases. Physically, higher values of $kH$ refers to the deep water case where waves are far away from the bottom and hence don't feel the bottom's effect, even though the resonance condition is satisfied, technically. Further, in a general setting where different interfaces are moving at different velocities, it may not be possible to form a polynomial equation in terms of intrinsic frequency. In any case, the energy of

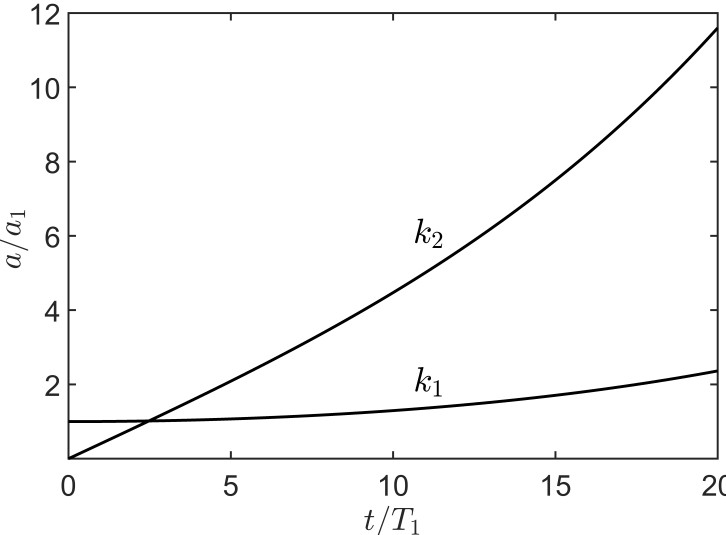

**Figure 3.** Growth of amplitudes of a wave on $\mathcal{SG}^-$ branch which resonates a wave on the same branch. Incident wave wavenumber is $k_1 H = 0.06$ while resonant wave has the wavenumber $k_r H = 3.13$. Other relevant parameters are $k_b H = 3.19$, $\omega_1/\sqrt{g/H} = -0.0213$, $\omega_2/\sqrt{g/H} = 0.0213$, $U/\sqrt{gH} = 0.70$, $N = 2048$, $M = 2$, $a_{1s}/H = 2.5 \times 10^{-5}$, $a_b/H = 0.025$, $T_1/\Delta T = 1024$. Here, '$N$' denotes the number of Fourier modes used in numerical simulation whereas '$M$' denotes the order of nonlinearity taken into account. (See Raj and Guha (2019) for details)

waves can always be found out and hence, the necessary condition for existence of an explosive Bragg pair is that a branch of the dispersion curve having negative energy must exist.

Finally, we point out that for case of explosive triads between three waves (discussed in Craik and Adam (1979)), the existence of negative energy didn't necessarily mean explosive interaction. The explosive instability occurs when out of the three waves involved, "the wave of greatest frequency has energy of opposite sign from the other". However, in this case of Bragg resonance, only two waves are involved with the third wave being the bottom ripple having zero frequency and the other two waves having same magnitude of frequency. Therefore, if the bottom is at rest, Bragg resonance involving waves having opposite energy will necessarily mean an explosive growth.

We have also numerically simulated a case of explosive instability using a HOS code (Alam et al., 2009b), which was extended by Raj and Guha (2019) to incorporate shear in it. Both the incident wave and the resonant waves are on $\mathcal{SG}^-$. The incident wave having wavenumber $k_1 H = 0.06$ interacted with the bottom ripples of wavenumber $k_b H = 3.19$ to resonate a wave having wavenumber $k_2 H = 3.13$. Other relevant parameters are mentioned in the caption of the figure. It can be seen that within 20 time periods, the amplitude of the resonant wave has become over 10 times that of the initial amplitude of the incident wave.

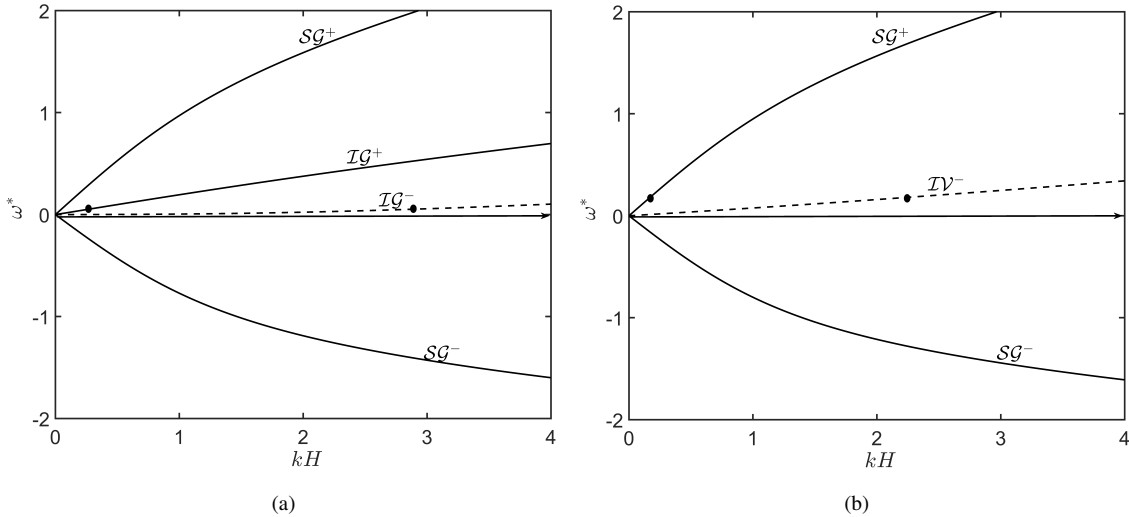

**Figure 4.** (a) Dispersion curves for a uniform current (flowing over a stationary bottom boundary). $U_u^* = U_l^* = U_b^* = 0.1, h_u/h_l = 1/3, \rho_u/\rho_l = 0.95$. (b) Dispersion curves for shear in bottom layer and single density fluid. $U_u^* = U_l^* = 0.1, U_b^* = 0, h_u/h_l = 1/3, \rho_u = \rho_l$. Dashed curves have negative energy. $U$ has been non-dimensionalised with $\sqrt{gH}$, where $H = h_u + h_l$

## 3  Explosive resonance in a two-layered flow

In this section, we consider a two-layered flow having a density $\rho_u$ and mean shear $\Omega_u$ in the upper layer and a density $\rho_l$ and mean shear $\Omega_l$ below it. By a two-layered flow we don't necessarily mean that there has to be a density difference between the two layers. We simply mean that there has to be either a shear jump or a density jump (or both) between the two layers,

which may lead to a perturbation vorticity generation at the interface. We have already discussed the theory of explosive Bragg resonance; this section mainly shows that in a two-layered setting achieving the explosive Bragg resonance needs significantly lesser velocity. In a two-layered density stratification without any velocity field, there exists four different modes of wave propagation- two surface modes and two interfacial modes propagating symmetrically in both directions. Whereas, for a one-layered setting, the energy transfer was limited to the surface only, in this case, waves on the pycnocline may also participate

in the energy exchange. The intrinsic frequency of a gravity wave on a pycnocline is significantly lesser than that of the surface because the density contrast at air-water interface is far more than the density contrast at the pycnocline. Therefore, the condition for explosive instability i.e. $|Uk| > |\omega_{in}|$ maybe satisfied at significantly lower values of $U$ and $k$ for the interfacial mode. As can be seen from figure 4(a), a uniform current having a small Froude number of 0.1 may also lead to the negative energy branch ($\mathcal{IG}^-$) and consequently, any Bragg triad involving negative energy branch will be prone to explosive growth.

One such Bragg triad has been shown using discs '•' in the figure.

Further, in cases where the density ratio limits towards one ($R \rightarrow 1$), there cannot exist a gravity wave at the interface. However, if there is a shear jump, then the interface may support a vorticity wave. Thus, there still may be energy transfer from the surface to the vorticity interface through Bragg resonance. The other way such an energy transfer can happen is through a

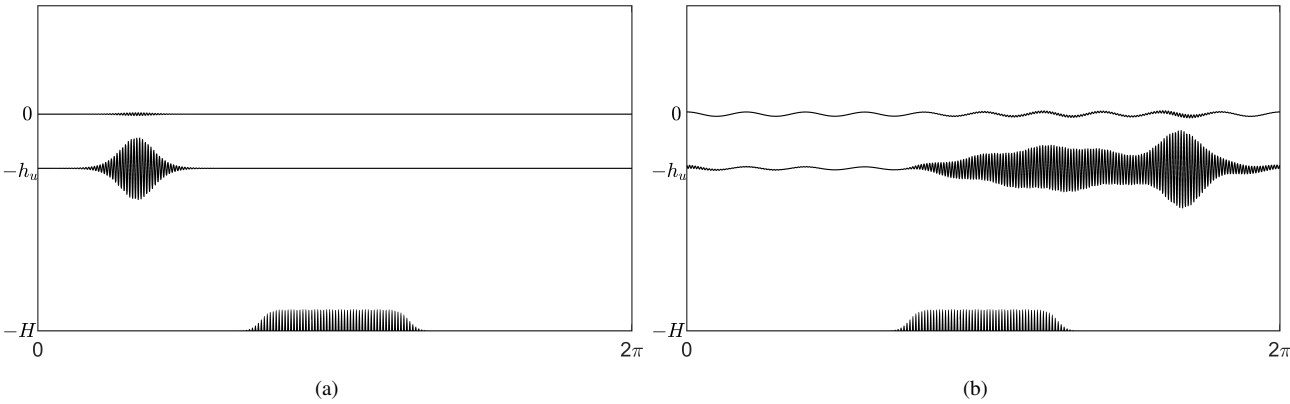

**Figure 5.** (a) Incident interfacial mode wave packet having $k_1 H = 2.1$ at $t = 0$. (b) The incident mode grows while simultaneously resonating with a surface mode having $k_2 H = 0.1$ Other relevant parameters are $k_b H = 2.0, U_u^* = U_l^* = 0.1565, U_b^* = 0, h_u/h_l = 1/3, \rho_u/\rho_l = 0.95,$ $\omega_1^* = \omega_2^* = 0.1095, N = 1024, M = 2, a_i = 0.01 h_u, a_b = 0.1H, T_1/\Delta T = 512$. $U$ has been non-dimensionalised with $\sqrt{gH}$, where $H = h_u + h_l$.

wave-triad interaction, which was the theme of the paper by Drivas and Wunsch (2016). For wave triad interaction involving a vorticity wave, no such explosive instability was reported by Drivas and Wunsch (2016) but for Bragg resonance, we find that the resonant interaction involving the vorticity wave are prone to explosive instability if shear is in bottom layer. For a piecewise linear velocity profile having $U_u = U_l = U > 0, U_b = 0$ for which $\Omega_u = 0, \Omega_l = U/h_l$, the dispersion relation is given by,

$$5 \quad \omega_{in}^3 + \frac{\Omega_l T_l}{(1 + T_u T_l)} \omega_{in}^2 - \frac{gk(T_u + T_l)}{(1 + T_u T_l)} \omega_{in} - \frac{gk\Omega_l T_u T_l}{(1 + T_u T_l)} = 0, \tag{3.20}$$

where, $T_u = \tanh kh_u$ and $T_l = \tanh kh_l$.

It can be easily seen from the above equation (3.20), which is cubic equation in $\omega_{in}$ that for $\Omega_l > 0$, that the product of the three roots is positive. Further, we know that one gravity wave branch has a positive intrinsic frequency while the other has negative intrinsic frequency. This basically means that the vorticity wave branch must have a negative intrinsic frequency. However, for any value of a positive velocity $U$, the frequency of vorticity waves becomes positive. The dispersion relation showing the negative energy branch ($\mathcal{IV}^-$) has been plotted in the figure 4(b).

In order to better visualize explosive resonance, we have also simulated an explosive case in a two-layered flow. The incident wave is a relatively short interfacial mode having a wavenumber $k_1 H = 2.1$ whereas the resonant wave is a surface mode having the wavenumber $k_2 H = 0.1$. The bottom ripple has a wavenumber $k_b H = 2.0$ and the velocity at the bottom is zero and the velocity both at the surface and the interface is $U/\sqrt{gH}$=0.1565. Figure 5(a) shows the wave packet at $t = 0$ and figure 5(b) is the wave packet at the final time. The resonant wave having large wavelength can also be observed.

Further, we have also plotted the variation of Fourier transform of the interface with time in figure 6. Increase in both the amplitudes pertaining to $k = k_1$ and $k = k_2$ shows 'explosive' growth.

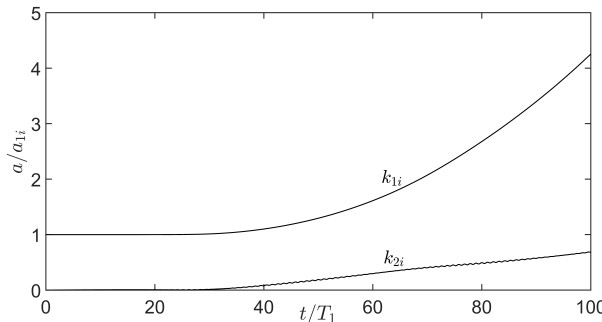

**Figure 6.** Growth of amplitudes of a wave on $\mathcal{IG}^+$ branch which resonates a wave on $\mathcal{SG}^+$ branch. Parameters are same as in figure5. The subscript '$i$' indicates that the amplitudes are measured at the pycnocline.

## 4 Conclusions

Bragg resonance has been traditionally understood in the absence of any background velocity field, and in such scenarios, the amplitude of one wave decays to cause a growth in the amplitude of the other wave. However, in the presence of a velocity field, we show that it is possible to have an exponential growth in the amplitudes of the waves. Notably, even though the presence of a velocity field is necessary, this exponential growth is not a consequence of linear instability because the velocity field chosen is not linearly unstable to perturbations. Further, this simultaneous growth happens without any violation of the law of conservation of energy similar to the explosive instability arising due to wave triad interaction (Craik and Adam, 1979).

We have explored the possibilities of explosive triads (i.e. all the waves involved in the system grows while keeping the energy conserved) where one of the involved waves is the bottom ripple. Although we have shown it for certain velocity profiles but the fundamental reason is the Doppler shift of the waves with respect to the bottom. Unlike wave triad interaction, where involvement of a negative energy wave *may* lead to explosive growth, in case of Bragg resonance, presence of such a wave *will* lead to explosive growth. For a single layered flow, the velocity required for the formation of an explosive triad is moderately high, however when pycnocline is present, explosive instabilities can occur even for small velocities. This is because of low intrinsic frequency of the interfacial gravity wave due to which even a low velocity can Doppler shift the intrinsic frequency to change the sign of the observed frequency.

*Author contributions.* The authors contributed equally.

*Acknowledgements.* A.G. thanks Alexander von Humboldt foundation for funding support.

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
