# Peer review of "Explosive instability due to flow over a rippled bottom"

_Nonlinear Processes in Geophysics, 2019_

## Referee Comment (RC1) · Anonymous Referee #1 · 26 Apr 2019

The manuscript at hand presents a theory about resonant triadic interactions for water waves propagating on top of a sinusoidal bottom, in the presence of a linearly sheared current. In the absence of current, this interaction is well known, and the only possible interaction involves a propagating wave, a reflected one, and the bottom undulations. In this work, the authors investigate the effect of a linearly sheared current and investigate the possible existence of an explosive instability (i.e. unboundedly growing with time). Indeed, since the dispersion equation admits new possible solutions, various triadic interactions are now made possible. At the end of the manuscript, a short discussion is introduced about this explosive instability in the presence of a two-layered flow, explaining this instability is more likely to occur in such configurations. My overall impression of the manuscript is not very good. It is unclear, and for this reason, the

point made by the authors is rather unconvincing.

The first reason concerns the way the dispersion equation is addressed. The authors present the roots of the dispersion equation as the intersection of the two branches named $SG^+$ and $SG^-$, but I could not find any definition of these two curves. As far as I understand, for each value of $k$, the authors find the two roots of equation 2.2, and then add $kU$ to each of these roots, obtaining the curves $SG^+$ and $SG^-$. Unfortunately, in that process, two other branches are neglected (the branches $\omega_{in} - kU$). Indeed, it is well known that even on a linearly sheared current, the dispersion equation admits four distinct solutions. This procedure has a strong impact on the following discussion, since the reader never realizes which wave is considered (as far as I understand, the counter propagating wave is always excluded, since the intrinsic frequency $\omega_{in} = \omega + kU$ is not even considered here). And this is not a minor remark. It is important, I believe, to understand which waves are considered and discussed here, by comparison with the classical four solutions of the dispersion equation (see e.g. the review paper by Peregrine). Anyway, a classical way to proceed with this difficulty is to consider both $k < 0$ and $k > 0$, and seek for positive values of $\omega$. I assume this modification would possibly impact the results presented here (possibly, equations 2.6 and 2.7 might be affected, and the overall discussion impacted.

The second point which remained dark concerns the obtention of equations 2.9. These equations are just dropped into the paper, without any reference, neither theoretical background. We only know that a time dependence is assumed for both a1 and a2, but this sounds very surprising. What exactly is the physical problem at hand? How is it possible that both these amplitudes suffer a time evolution simultaneously? Aren't we considering forcing by an incident wave (which would certainly have a constant amplitude)? This remark is also important. It is almost impossible for the reader to reach a clear idea of the problem considered, including its boundary conditions, or its spatial extension.

These two points seem of major importance to me, and the impression of unclarity

remains valid all along the manuscript. Therefore, I cannot recommend this manuscript for publication in its present form.

---

## Referee Comment (RC2) · Anonymous Referee #2 · 5 May 2019

This manuscript presents a theoretical study of the two-dimensional "explosive instability" of resonant triads between two surface waves and either a bottom ripple or a vorticity wave on a velocity interface. I believe that this might ultimately be publishable in Nonlinear Processes in Geophysics, but not in its present form. The two principle reasons are listed below.

1. I am concerned that the results from section 2 (two-layered fluid) are not sufficiently novel. The explosive instability in a two-layer fluid using piece-wise linear velocity profiles (analogous to the present work) was previously investigated by Voronovich and Rybak (Oceanology 1977) and Voronovich et al. (Isvestia 1980). The work of Alam (JFM 2012) on triad resonance between two surface waves and an interfacial wave (without shear, and hence without the explosive instability) should also be recognized.

The results in the present manuscript appear to be very similar to these prior studies. The authors need to revise the manuscript to acknowledge this and discuss the differences between the results, or else remove this section if the results are essentially identical.

2. On page 8, the authors refer to a numerical simulation of the explosive instability for a bottom ripple, but present no results. Instead, an archive pre-print is referenced. I recommend incorporating the numerical simulations into a revised manuscript, as this would make the article much more complete and thorough. Readers would benefit from seeing illustrations of the flow showing how the explosive instability works in practice. As a further suggestion, a numerical simulation of the two-layer case (Sect. 2) would also be a valuable addition, as this could establish the existence of an explosive instability with velocity profiles that are more realistic than the piece-wise linear profiles used here. Such a simulation could be publishable even if the theoretical results are essentially identical to the previous work of Voronovich.

---

## Author Comment (AC1) · 16 May 2019

We thank you for your useful comments, suggestions and constructive criticisms, which have significantly helped us in improving the paper. We have split your comments into several parts (marked in red), so that all the queries could be answered in a point-wise manner. Responses to your comments are marked in red in the paper.

**Opening comments:** The manuscript at hand presents a theory about resonant triadic interactions for water waves propagating on top of a sinusoidal bottom, in the presence of a linearly sheared current. In the absence of current, this interaction is well known, and the only possible interaction involves a propagating wave, a reflected one, and

the bottom undulations. In this work, the authors investigate the effect of a linearly sheared current and investigate the possible existence of an explosive instability (i.e. unboundedly growing with time). Indeed, since the dispersion equation admits new possible solutions, various triadic interactions are now made possible. At the end of the manuscript, a short discussion is introduced about this explosive instability in the presence of a two-layered flow, explaining this instability is more likely to occur in such configurations. My overall impression of the manuscript is not very good. It is unclear, and for this reason, the point made by the authors is rather unconvincing.

**(1)** The first reason concerns the way the dispersion equation is addressed. The authors present the roots of the dispersion equation as the intersection of the two branches named SG+ and SG−, but I could not find any definition of these two curves. As far as I understand, for each value of k, the authors find the two roots of equation 2.2, and then add kU to each of these roots, obtaining the curves SG+ and SG−. Unfortunately, in that process, two other branches are neglected (the branches $\omega_i n - kU$). Indeed, it is well known that even on a linearly sheared current, the dispersion equation admits four distinct solutions. This procedure has a strong impact on the following discussion, since the reader never realizes which wave is considered (as far as I understand, the counter propagating wave is always excluded, since the intrinsic frequency $\omega_i n = \omega + kU$ is not even considered here). And this is not a minor remark. It is important, I believe, to understand which waves are considered and discussed here, by comparison with the classical four solutions of the dispersion equation (see e.g. the review paper by Peregrine). Anyway, a classical way to proceed with this difficulty is to consider both $k < 0$ and $k > 0$, and seek for positive values of $\omega$. I assume this modification would possibly impact the results presented here (possibly, equations 2.6 and 2.7 might be affected, and the overall discussion impacted.

**Our response:** Kindly note that we have actually mentioned in the paper what $\mathcal{SG}^+$

and $\mathcal{SG}^-$. It appears below Eq. (2.2): "*The positive intrinsic frequency branch has been labeled as $\mathcal{SG}^+$, while the negative one as $\mathcal{SG}^-$*." We have assumed $k > 0$ without loss of generality, as apparent from Fig 2. The convention $\omega > 0$ could also be taken, but have chosen the former.

The dispersion relation obtained for waves on a sheared flow is given as

$$\omega_{in}^2 + \Omega \tanh(kH)\omega_{in} - gk \tanh(kH) = 0,$$

where, $\omega_{in} \equiv \omega - Uk$. In other form, this can be written as,

$(\omega - Uk)^2 + (\omega - Uk)\Omega - gk \tanh kH = 0$. This is an exact dispersion relation with no approximation or assumptions. It can be observed that if $(\omega_0, k_0)$ is a solution of above equation, then $(-\omega_0, -k_0)$ will also be a solution. In other words, the dispersion relation is symmetric about the origin. If we plot this dispersion relation, allowing $k$ and $\omega$ to take any real value, then the dispersion relation will have 4 branches: $(|\omega|, |k|)$, $(|\omega|, -|k|),(-|\omega|, |k|)$ and $(-|\omega|, -|k|)$. As mentioned above, the two branches in the $k < 0$ plane are simply the reflection of branches in the $k > 0$ plane about the origin. We therefore can ignore one half of the dispersion curve which lies on $k < 0$ plane. Thus both $(|\omega|, |k|)$ and $(-|\omega|, -|k|)$ denote waves propagating along the positive $x$ direction (+ve phase speed) and both $(|\omega|, -|k|)$ and $(-|\omega|, |k|)$ denote waves propagating along the negative $x$ direction (-ve phase speed).

You have correctly mentioned that "Anyway, a classical way to proceed with this difficulty is to consider both $k < 0$ and $k > 0$, and seek for positive values of $\omega$". We have done a very similar thing but in a different way. Rather than considering both $k < 0$ and $k > 0$ and fixing $\omega > 0$, we have fixed $k > 0$ and have considered both $\omega > 0$ and $\omega < 0$. In the convention suggested by you, $k < 0$ will denote counter-propating wave while in the convention used by us, $\omega < 0$ denotes the counter-propagating wave. We hope that the confusion regarding the convention is now addressed.

We also add that it is very common practice to consider only $k > 0$ plane while studying

the dispersion curves. See for example, all the dispersion curves in Cairns (1979), Carpenter et al. (2011). However, just to clarify this point, in the revised version, we have added the following sentence after equation (2.2):

For this paper, without a loss of generality, we have considered a positive $k$ and have allowed the frequency $\omega$ to be either positive or negative. As a consequence, $\omega > 0$ means a positively travelling wave while $\omega < 0$ means a negatively travelling wave in the stationary reference frame.

**(2)** The second point which remained dark concerns the obtention of equations 2.9. These equations are just dropped into the paper, without any reference, neither theoretical background.

**Our response:** The general traid formulation is well known and can be found, for example, in Craik (1988). In this paper, we are investigating a special kind of triad - the Bragg triad, where one of the constituent 'waves' is the bottom ripple. The details of the derivation has been provided in our recent paper, Raj and Guha (2019). We have made some small rewordings around equation (2.9) so that the reader knows where to find the Bragg resonance equations. We hope this would avoid both confusion as well as repitition.

We only know that a time dependence is assumed for both $a_1$ and $a_2$, but this sounds very surprising. What exactly is the physical problem at hand?

The physical problem at hand is that two waves are passing over a bottom topography and both of them are exchanging energy with each other via the mediation of the bottom ripples (as if the bottom ripple is a constant amplitude, stationary wave which forms a triad with the two real waves). In doing so, the amplitude of one of the constituent wave usually increases and that of the other wave decreases. However, for

the particular case of 'explosive growth', the amplitudes of both the waves increase. Apparently it may appear as a violation of the conservation of energy, but it isn't; see Craik and Adam (1979). For clarity, we have included a sentence in the last paragraph of the first section:

Here, the system consists of a surface wave propagating over a rippled bottom topography and the fluid flowing with a mean velocity profile as shown in figure 1(a). For explosive instability, we expect this wave to grow along with its explosive counterpart while simultaneously conserving the energy of the system.

How is it possible that both these amplitudes suffer a time evolution simultaneously? Aren't we considering forcing by an incident wave (which would certainly have a constant amplitude)?

No, there is no other incident wave of constant amplitude. Firstly, it is well known that **three** different waves can exchange energy with each other with amplitude of all the waves changing (known as wave triad interaction; see Craik (1988)). Secondly, it is also known that there can be a case that **three waves interact but amplitude of one stays constant** and other two register an exponential growth (known as parametric subharmonic instability (PSI)). This is nothing but a subcase of the first case i.e. wave triad interaction. Thirdly, it is also known that only **two** different waves can exchange energy with each other provided that there is a bottom ripple present (Bragg resonance). Our paper is a sub-case of the third case in which the two waves interact with the bottom and both the waves register a growth.

This remark is also important. It is almost impossible for the reader to reach a clear idea of the problem considered, including its boundary conditions, or its spatial extension. These two points seem of major importance to me, and the impression of unclarity remains valid all along the manuscript. Therefore, I cannot recommend this manuscript for publication in its present form.

We hope that now we have been able to clarify your doubts.

[revised manuscript text omitted]

---

## Author Response (AR1)

**Comments to referee #2**

Raunak Raj & Anirban Guha

June 6, 2019

We thank you for your useful comments, suggestions and constructive criticisms, which have significantly helped us in improving the paper. We have split your comments into several parts (marked in blue), so that all the queries could be answered in a point-wise manner. Responses to your comments are marked in blue in the paper.

This manuscript presents a theoretical study of the two-dimensional "explosive instability" of resonant triads between two surface waves and either a bottom ripple or a vorticity wave on a velocity interface.

In this paper, one 'wave' has to be the bottom ripple. Hence, the resonance is bewteen two surface/interface gravity or vorticity or vorticity–gravity wave along with the bottom ripples.

I believe that this might ultimately be publishable in Nonlinear Processes in Geophysics, but not in its present form. The two principle reasons are listed below.

1. I am concerned that the results from section 2 (two-layered fluid) are not sufficiently novel. The explosive instability in a two-layer fluid using piece-wise linear velocity pro-files (analogous to the present work) was previously investigated by Voronovich and Rybak (Oceanology 1977) and Voronovich et al. (Isvestia 1980). The work of Alam (JFM 2012) on triad resonance between two surface waves and an interfacial wave (without shear, and hence without the explosive instability) should also be recognized. The results in the present manuscript appear to be very similar to these prior studies. The authors need to revise the manuscript to acknowledge this and discuss the differences between the results, or else remove this section if the results are essentially identical.

Most of the theoretical work of our paper is contained in the section 2 itself (i.e. the single–layered fluid). The purpose of the section concerned with two–layered fluid (section 3) is mainly to highlight that in a two–layered flow, the condition of explosive instability is met at considerably lower velocity. We believe that we have adequately explained in the section that why a two–layered flow with small density jump may easily give rise to explosive instability.

Further, we have cited the papers by Voronovich and Alam, as suggested by you, in the revised version of the paper.

2. On page 8, the authors refer to a numerical simulation of the explosive instability for a bottom ripple, but present no results. Instead, an archive pre-print is referenced. I recommend incorporating the numerical simulations into a revised manuscript, as this would make the article much more complete and thorough. Readers would benefit from seeing illustrations of the flow showing how the explosive instability works in practice. As a further suggestion, a numerical simulation of the two-layer case (Sect. 2) would also be a valuable addition, as this could establish the existence of an explosive instability with velocity profiles that are more realistic than the piece-wise linear profiles used here. Such a simulation could be publishable even if the theoretical results are essentially identical to the previous work of Voronovich.

Firstly, we have not explained the numerical method in detail in this paper. The paper concerned with details of numerical method is the arXiv pre-print which is now published in JFM and we have revised the citation. Secondly, we had included the results of one simulation pertaining to single-layered fluid in the first version (Figure 3). The figure illustrates that the amplitude of both of the waves (i.e. the spatial Fourier transform of the waves at different times) grow exponentially. However, we did not include that how the waves and explosive instability look like in the real space. Therefore, thirdly, in the revised version of the paper, we have added three new figures showing that how the explosive instability may look like in the real space (Figure 5). Fourthly, we would like to clarify that the numerical simulation too use a piece-wise linear velocity profile. The numerical simulation is based on the method described in Alam et. al. (2009, Paper II). We have extended the method to include a piecewise linear velocity profile in Raj & Guha (2019). Because the method is based on the potential flow theory, it will not be possible to use a fully non-linear velocity profile (i.e. profile that is continuously differentiable).

We thank you for your suggestion which has helped us in enhancing the novelty of the paper. We hope that now we have been able to clarify your doubts.

**References**

R. A. Cairns. The role of negative energy waves in some instabilities of parallel flows. *J. Fluid Mech.*, 92(1):1–14, 1979.

J. R. Carpenter, E. W. Tedford, E. Heifetz, and G. A. Lawrence. Instability in stratified shear flow: Review of a physical interpretation based on interacting waves. *Appl. Mech. Rev.*, 64(6): 060801, 2011.

A. D. D. Craik. *Wave Interactions and Fluid Flows*. Cambridge University Press, July 1988.

A.D.D. Craik and J.A. Adam. 'Explosive' resonant wave interactions in a three-layer fluid flow. *J. Fluid Mech.*, 92(1):15–33, 1979.

R. Raj and A. Guha. On bragg resonances and wave triad interactions in two-layered shear flows. *J. Fluid Mech.*, 867:482–515, 2019.

---

## Author Response (AR2)

**Comments to the Associate Editor**

Raunak Raj & Anirban Guha

July 21, 2019

Dear Prof. Grimshaw

Thank you for the valuable comments and constructive criticisms. We have tried to take all of these into consideration while revising our paper. Your comments appear in blue and our replies in black.

Referee 2 has recommended accept as is, but referee 1 has said the presentation remains confusing, and needs further editing. I agree, and ask that you work over the text and in particular avoid repitition.

We have tried to make minor edits in the paper (since we are not sure exactly which parts of the presenation referee 1 found confusing.

I would contest the statement that explosive instability can arise even with a uniform flow, as any unirom flow can be removed by a Galilean transformation

We would politely disagree with this comment. A uniform flow creates a velocity difference between the fluid and the bottom (which is at rest). Thus, it is not merely an absolute velocity $U$ of the fluid, but it is also the relative velocity $\Delta U = U$ between the bottom ripple and the fluid. Galilean transformation cannot remove any relative velocity. In other words, if we start moving with a velocity $U$, then yes, we will see the fluid at rest. But, we will also see the bottom ripples moving with a velocity $-U$. So, even though the flow is uniform, even a uniform flow will create a velocity difference between a bottom ripple and the waves in the fluid (say at the surface). As we have already discussed in the paper, this instability deals with the situation when one of the 'waves' in the system involved is the bottom ripple.

It might also be noted, if the bottom ripple was not involved i.e. if this was an instability between three 'actual' waves, then a uniform flow wouldn't make a difference.

Finally, in the paper, we had also mentioned the reason briefly-

"We emphasize here that, although the actual velocity profile is of some relevance, what matters the most is the Doppler shift between the bottom ripples and the surface. Therefore,even a uniform current would have worked just fine."

Also I would that negative energy concepts are best understood using wave action and averaged Lagrangians, which makes it clear that negative energy waves can only arise when there is a background flow. I take the liberty of referring to my review article on this, which gives the key references, Ann. Rev. Fluid Mech., vol 16, 1986, 11-44

Thank you for the explanation and the reference. We have added it in the text and it is colored in red.

Sincerely

Raunak and Anirban.

---

## Author Response (AR3)

**Comments to the Associate Editor**

Raunak Raj & Anirban Guha

July 22, 2019

We are nearly there, but I see from your response that further changes are needed. The confusion is your use of the term "uniform current" eg pm l 17, p2, l 15 p3, and Figure 4 caption. For most readers, "uniform current" means constant in both vertical and horizontal. This is not what you mean, and so a different phrase is needed.

Dear Prof. Grimshaw

Thank you very much for this comment. We see what lead to your confusion in the first place. Now we have changed the confusing definition to 'uniform current (flowing over a stationary bottom boundary)' or 'uniform mean flow (flowing over a stationary bottom boundary)'. We hope that now the issue is resolved.

Sincerely

Raunak and Anirban.